# Healthcare Professionals’ Experiences of Brief Admission by Self-Referral for Adolescents with Self-Harm at Risk of Suicide—A Qualitative Interview Study

**DOI:** 10.3390/bs15091210

**Published:** 2025-09-05

**Authors:** Rose-Marie Lindkvist, Kajsa Landgren, Sophia Eberhard, Björn Axel Johansson, Olof Rask, Sofie Westling

**Affiliations:** 1Department of Psychiatry, Skåne University Hospital, 221 85 Lund, Sweden; 2Psychiatry, Department of Clinical Sciences Malmö, Lund University, 221 00 Lund, Sweden; 3Psychiatry, Habilitation and Aid, Child and Adolescent Psychiatry, Regional Inpatient Care, Emergency Unit, Region Skåne, 205 02 Malmö, Sweden; 4Department of Clinical Sciences Lund, Division of Child & Adolescent Psychiatry, Lund University, 221 85 Lund, Sweden

**Keywords:** brief admission, child & adolescent psychiatry, experiences, healthcare professionals, non-suicidal self-injury, prevention, self-harm, self-referral, suicide attempt, suicide ideation

## Abstract

Brief Admission by Self-referral (BA), a standardized crisis intervention for individuals with repeated self-harm or suicidal behavior, was adapted for adolescents from 13 years in Region Skåne, Sweden, in 2018. BA aims to offer access to support based on autonomy and has been associated with reduced need of emergency care. Interviews with adolescents and legal guardians have pointed to BA as valuable and challenging, and professional support as key. This study aims to describe healthcare professionals’ (HCPs) experiences of BA for adolescents with self-harm at risk of suicide. Interviews six years after implementation with fourteen HCPs from outpatient and inpatient psychiatric care were analyzed with qualitative content analysis. BA was perceived as valuable caretaking without taking over, promoting mental growth and agency by being brief and granting access. It was described as offering relief to families and HCPs, although perceived to lack a sufficient level of legal guardian participation. Key work processes included being grounded in leadership and outpatient treatment. Challenges included system inflexibility and fitting BA into the physical care context. The results of this study may support future implementation of BA for adolescents with self-harm at risk of suicide and add guidance around potential pitfalls.

## 1. Introduction

Self-harm in adolescence, with or without suicidal intent is a global health problem ([6]). In the general population, the prevalence of self-harm increases in early adolescence, peaks between 15 and 17 years of age, and levels off in the transition to adulthood ([25]). Two percent of children under 13 years have attempted suicide ([20]) and nine percent of individuals between 14 and 21 years ([33]). Non-suicidal self-harm has been reported by 16 percent of adolescents ([2]), by six percent of children under 13 years ([20]) and by 23 percent of adolescents between 12 and 18 years ([3]). Self-harm is more common among young people in inpatient care where the prevalence of suicide attempts is around 12 percent ([20]). There is a continuous need for initiatives for prevention and management directed at adolescents with self-harm and suicidal ideation.

In 2018 Child- and Adolescent Psychiatry (CAP) in the region Skåne in Sweden implemented Brief Admission by self-referral (BA) for adolescents 13–17 years old with self-harm at risk for suicide ([9]). The initiative was inspired by positive experiences from adult psychiatric care ([12]). The aim was to better meet needs to reduce suicide and self-harm among adolescents and encourage their autonomy with respect to psychiatric care, at times when ongoing outpatient care services may be insufficient. BA was aimed at adolescents with marked instability, meeting at least three of the nine diagnostic criteria for borderline personality disorder, for whom psychiatric emergency admissions have questionable effects on repetition of self-harm and the risk for suicide ([6]; [32]).

Unlike other psychiatric admissions BA, a crisis management tool, does not require an initial psychiatric assessment by a physician, deciding whether the adolescent should be admitted or not. Instead, it is up to the adolescent with access to BA, secured through a specific process including a contract negotiation, to initiate an admission, which may last up to three nights, maximum three times per month. The set-up of BA requires responsibility and active participation of the adolescents and offers freedom in relation to when and how to use it. Support during admissions is delivered by healthcare professionals (HCPs), educated in nursing and in the BA-approach characterized by warmth, validation and respect and who are to care for adolescents according to personalized needs. Ongoing treatment in psychiatric outpatient care is a prerequisite for access to BA. Legal guardians are involved in the contract negations and are welcome to join adolescents during admissions provided that the adolescents want to. Legal guardians are contacted over telephone if they are not physically present during BA admittance or discharge. Besides having access to BA, the adolescent might need and is entitled to all other necessary psychiatric treatment as usual.

Observational data has indicated decreases in emergency care visits, emergency admissions and inpatient days among adolescents with self-harm at risk of suicide when getting access to BA ([8]). In qualitative interviews, adolescents have described BA as a safe place for recovery, as demanding, self-motivating, and available, appreciating as well as doubting the trust they were given through the BA contract ([19]). Legal guardians, who are, in contrast to emergency admissions, usually not admitted with adolescents on BA, have shared experiences of BA as both a gift of safety, well-being and liberation, and as a robbery, where they felt out of control during admissions perceived to be without purpose or plan ([11]). Prior studies point towards the importance of outpatient and inpatient care HCP roles in relation to BA and the need to understand perspectives of BA among those targeted and involved, as BA stands out compared to other psychiatric admissions. Qualitative research on HCPs experiences of BA within adult psychiatry has indicated perceptions of an increased sense of safety, improved collaborations and positive effects of focusing more on recovery ([1]; [15]; [17]). HCPs satisfaction with BA for adolescents has been indicated as high in relation to implementation, acceptance of the BA model and in terms of perceived benefits for adolescents ([9]).

This study aimed to expand on HCPs perspectives on BA for adolescents with self-harm at risk of suicide based on in-depth qualitative interviews regarding perceived effects and mechanisms of BA, key work processes, challenges and suggestions for improvement and development six years after implementation.

## 2. Materials and Methods

### 2.1. Design

This was a descriptive inductive qualitative study ([27]) on HCPs experiences of BA in CAP care in Skåne, Sweden. Subjective experiences shared in individual interviews aimed to allow for complex meanings. Participants’ perspectives were the results of interactions between the interviewed participants and the researcher and with consideration of the context where the participants worked (outpatient or inpatient CAP).

### 2.2. Setting

The study was conducted in October and November in 2024 within CAP care in the region Skåne, Sweden. Sweden is divided into 21 regions with Skåne as the country’s southernmost region with 1,400,000 residents, of whom 300,000 are aged 0–17 years, including 90,000 teenagers, the age group most common in CAP inpatient care. The region’s only CAP inpatient unit, serving the region’s 13 CAP outpatient units, had an emergency 24/7-unit and two inpatient care units, including two beds dedicated for BA, located near a unit for treatment of eating disorders. At the 24/7 emergency and inpatient care unit in Malmö worked registered nurses and licensed practical nurses specialized in psychiatry, whereof two with extensive experience of working with adolescents with self-harm and suicidal behavior who were dedicated to coordinate BA. Other HCPs working at the inpatient care unit included family therapists, senior consultants in child and adolescent psychiatry, psychologists, counselors, secretaries, one physiotherapist, one occupational therapist, one pharmacist, one music therapist, and, in periods, residents and interns.

At the time of the study, 40 adolescents had an active BA contract and close to 200 BA contracts had been negotiated with adolescents since implementation in 2018. Over the prior five years, all CAP outpatient units in the region had at least one adolescent with a BA contract, indicating that the method was well-established in the region. According to statistics gathered at the BA unit, around 60 percent of adolescents with a BA contract were using it by actively self-admitting.

### 2.3. BA for Adolescents

The structure of being able to self-refer to BA according to individual needs of crisis management closely followed the manual for BA for adults ([12]). Differences compared to the adult version of BA, for which it was originally developed, were medications being handled by registered nurses, cigarettes, lighters, etc., were controlled by HCPs, the possibility of requesting admission to BA was limited to daytime, involvement of legal guardians or other caretakers in the contract negotiation which took place every 6 months, and informing legal guardians about admission and discharge. Adolescents aged 13–17 years were eligible for BA if they had been treated in emergency care during the last six months and had a complex psychiatric condition with features of emotional instability, including self-harm, and suicidality. In addition, adolescents were required to receive structured ongoing outpatient care directed at emotion regulation and anxiety management, such as dialectic behavioral therapy (DBT). Adolescents who were considered unable to understand the implications of BA, who had intellectual disabilities, or psychotic syndromes, or were cared for in state institutions for legal reasons or placed in emergency homes were not offered access to BA ([9]).

### 2.4. HCPs Roles in Relation to BA for Adolescents

BA is designed as a structured collaboration between inpatient and outpatient care. An overview of key roles and responsibilities among HCPs working with BA is presented in Table 1.

### 2.5. Participants

Potential study participants were recruited among HCPs working in the inpatient unit in Malmö and in outpatient units in the region. Participants were recruited through a three-step process using purposeful snowball sampling. Initially, nine HCPs who had been interviewed about their experiences of BA for adolescents for a non-published student paper were contacted. Next, emails were sent out to all HCPs in the region within CAP care (949 individuals) with written information about the study and a request to interested individuals with experience of working with BA for adolescents to volunteer to participate. Thirdly, those who were e-mailed were also asked about any suggestions that they might have of individuals they thought could be relevant to interview and asked to pass study information on to those. Those interested were contacted and informed about the study in more detail orally and if they were still interested a time for interview was booked. Recruitment resulted in 14 participants working in both outpatient and inpatient care, see Table 2 for an overview of participant characteristics.

### 2.6. Data Collection

Interviews, performed by the first author, opened with the question ‘Can you describe your role in relation to BA?’. Then participants were followed in their descriptions with follow-up questions based on answers to gain depths, such as ‘Tell me more about that’ or ‘What does it mean when you say…?’. A semi-structured interview guide was used containing open-ended questions related to perceived effects and mechanisms of BA, key work processes, important learnings, benefits and hurdles, challenges and suggestions for improvement and development of BA for adolescents. The order of subjects differed between interviews as the interviewer aspired to follow the participant and ask questions in an order which seemed most natural to avoid interruption and follow the participants’ chain of thoughts. The interview guide was developed in collaboration between all authors, inspired by previous research on BA ([11]; [19]; [17]), and reviewed by a representative for the non-governmental organization Swedish Partnership for Mental Health (NSPH), working for increased participation in healthcare among patients, users and family carers. The guide included examples of questions and a supportive table of potentially relevant subjects relating to aspects of freedom, safety, recovery, care environment, collaboration, routines and information. Out of the fourteen, nine interviews were digital (eight via zoom and one over telephone) and five interviews took place in physical meetings at the participants’ workplaces. Interviews lasted 25–70 min (mean 49 min).

### 2.7. Analysis

Interviews were audio-recorded, transcribed verbatim with the support of the Open AI tool Whisper (OpenAI, San Francisco, CA, USA), and then reviewed by R.-M.L., by listening through all interviews and correcting any errors in the transcriptions. Analysis followed the process of qualitative content analysis, where the text was divided into meaning units, which were condensed, coded and grouped into sub-themes and themes in order to process and structure content according to meaning ([4]; [5]; [13]). Analysis was conducted in a dynamic process led by R.-M.L. with support from K.L., both experienced in qualitative research in the field. They thoroughly read through interviews and discussed the content on several occasions throughout analysis. Meaning units were identified in a row-by-row division of the entire transcribed data set based on similar content and context. In some cases, meaning units consisted of a few words and in other cases an entire paragraph, depending on content density. The division of the text into meaning units, condensation and coding was initially performed by R.-M.L. and reviewed by K.L. who gave comments and suggestions in writing and orally, which were considered and discussed in joint meetings. R.-M.L. suggested groups of codes into sub-themes and themes which were discussed in several meetings between R.-M.L. and K.L. Analysis was performed at both manifest and latent level depending on perceived depth, where some content was coded close to the original wording of the participant based on visible and obvious components while other content was allowing for an in-depth analysis and abstracted according to the interpreted underlying meaning. R.-M.L. and K.L. put together results jointly whereafter results were shared and discussed with all authors.

### 2.8. Ethical Considerations

The study was granted ethical approval in Sweden (2020-01840; 2020-06-02). Participants were given written and oral information, including that participation was voluntary, were given the opportunity to ask questions and gave their consent to participate before the start of the interview. A code list was constructed to link each interview with a participant and stored separately from the data. Authors who knew the identity of the study participants (R.-M.L., K.L. and S.W.) were not involved in CAP care. AI-transcription was performed in line with the General Data Protection Regulation without internet connection when handling audio recorded interviews.

## 3. Results

The analysis resulted in themes and subthemes, see Table 3, presented with descriptions and quotations in text below.

### 3.1. Caretaking Without Taking over

#### 3.1.1. Promoting Mental Growth and Agency

Participants described that they had become enthusiastic over the concept of BA, providing ground for HCP growth, with HCPs being prepared to receive adolescents as they arrived. Meeting the adolescents with BA was phrased as a sort of advanced support service creating calmness and protection against self-harm impulses based on adolescents being listened to and taken seriously. Participants experienced that adolescents were able to take control over situations and be more independent. Where their emotions otherwise would be out of control, the adolescents had gained control through self-reflection when deciding on admission length, needs and purpose of BA with daily plans, whether it was to be activated through BA or to rest, go outside to meet friends or attend school. Participants said that BA contributed to responsibility and participation, allowing adolescents to tailor BA according to their individual needs, which had made an impression on the participants. They mentioned examples such as being surprised that the most common goal of BA was to manage school and that adolescents who did not seem to be ready could grow into BA.

According to the participants’ experiences BA increased adolescents’ confidence and well-being through control and consideration. Adolescents seemed to be motivated by being the pilot in their own care characterized by freedom and responsibility, according to participants. Participants, having the drive to increase autonomy for those who had been worse off during emergency admissions, emphasized the joy of seeing the effects of giving adolescents the power to choose admission. They said that it was hopeful to see adolescents who had demanded extensive support to be kept together, growing in independence and motivation with the help of BA. An example was how adolescents during contract negotiations were stimulated to formulate their strategies to handle anxiety in a future crise.


*“To be in control. To see that you can actually control your life. Because I think many of them have such a difficult time with their emotions and are like a leaf in the wind. They are exposed to situations where there is not much consideration. Here they can, at least for the moment, become the pilot themselves. And I think that motivates them into their work too.”*

*(interview 13)*



*“During contract negotiations, I have become better over time at emphasizing that the adolescent is the agent. You are. This is your contract. And this is important: I emphasize that the guardian cannot pressure you, do not get to decide when you should use BA. It is up to you. It is your choice. It is your opportunity to take control of yourself and the care even though you are 14 or 15 or 16, it doesn’t matter.”*

*(interview 7)*


Another important aspect of BA brought forward by the participants was the adolescents’ right to decide when they no longer needed BA, emphasizing the luxury and meaning of voluntarily ending their BA contract, while their legal guardians were listening and where the adolescents themselves could reflect and decide. Participants said that this was quite significant and unusual as many other things within psychiatric care were decided on over the adolescents’ heads.


*“It’s really nice for them to be able to say: no, I don’t think I need/BA/anymore. Because then they’ve reached a point where they’ve matured, or they’re not as broken anymore. And they dare to go out into life without this lifeline. (…) So, it’s kind of a triumph (…) That they choose this healthy way and want to be able to manage without it.”*

*(interview 12)*


#### 3.1.2. Being Brief

In relation to long, often several months of hospitalizations, BA had become a new way of contributing to reduced compulsory care and in-patient psychiatric care through the calming effects of brief admissions, well-adapted to needs. Although necessary during the worst times of suicidal risks, participants described how hospitalizations following emergency admissions could be devastating and even dangerous. Participants shared experiences of having felt that they had been groping in the dark on how to best help adolescents at risk of suicide when seeing self-harm episodes increase over time. In relation to this, participants talked about the value of the briefness of BA, being limited to three days. They said that this set-up fit well with their experiences of how processes leading to self-harm were typically rapid and that adolescents were not benefiting from being stuck in lengthy admissions.


*“Sometimes emotions can calm down quite quickly. (…) To prevent self-harm and to prevent a suicide attempt. And when that’s done there is no point in keeping going (…) And then you could go out again and feel that now I’ve slept, now I’ve eaten, now I’m recovered. Now I can manage again.”*

*(interview 3)*


Participants were describing BA as a minibreak from hard times with available HCPs. They described that they perceived that the adolescents appreciated to be able to come and land when it was difficult at home, and that the limited number of days was perceived to support reflection and responsibility.

#### 3.1.3. Granting Access

Participants emphasized the perceived benefit and the importance of the BA contract offering valuable availability. According to their experiences many adolescents tried BA a few times, especially in the beginning, to test if it was true according to what had been promised during the contract negotiation but then tended to use it less. Participants said that many adolescents were not using BA at all but still wanted to renew the contract at the biannual revision. Participants emphasized that it was important that adolescents knew that they could renew the contract, regardless of how they had used it to date. They said that just because adolescents were not using BA actively did not mean that the BA contract was not effective, because having the possibility to use it did more than what could be measured.


*“This document. It helped her to… catch herself in the moment and find other strategies instead. And be able to stay at home. That was safety for her. (…) She was admitted to compulsory care when she was 13 or 14. And when she was granted leave, she went straight out and cut up her arms. (Interviewer: Mm. Yes. And did she use BA later on?) No, but you can be sure that she kept her contract. And we have renewed it three times.”*

*(interview 1)*


Instead of having things just happening to the adolescents, BA provided an alternative to self-harm in situations which increased risks of impulsive behavior, such as sleep/food deprivation, school stress, relational problems or depression. Allowing adolescents to be able to call during times of instability has reduced the need for emergency care and long in-patient stays, according to experience. Participants emphasized the value of making it possible for adolescents to be welcome to access care quickly without having to wait for an emergency assessment. They said that BA offered a lifeline for a group who previously had been subjected to rejections, unpredictability and argumentation at the emergency room.


*“Finally, someone had understood what rejection does to this group. (…) It meant a lot more respect for the patient and the patient’s difficulties. Instead of us in psychiatry being some kind of gatekeepers and in charge.”*

*(interview 6)*


BA had become part of the crisis plan at times when DBT and learnt skills were not enough. Participants described that they experienced BA as something that could be kept in the back of the adolescents’ minds in critical situations. They could talk about BA with the adolescents, and the adolescents could choose BA instead of doing something self-destructive. Participants working at the inpatient care unit providing BA said that they were grateful to be able to listen and offer help proactively, instead of leaving legal guardians responsible.


*“The experience is that they… not just lay it on someone else, whether it’s parents or other staff or something. Because/they/know that/they/have these days available. And that/they/get to be a little mindful of how/they/spend them. In some way that tells/them/that/they/need to reflect and check in with/themselves/, before using/BA/.”*

*(interview 4)*


#### 3.1.4. Offering Relief to the System

Participants shared experiences of how BA was offering relief, peace and quiet to the families of adolescents by cooling down difficult and painful situations. BA was experienced to offer safety, hope, stress reduction, breathing space, contact and a pat on the back. Participants described perceived reduced risks of escalation and illness, because legal guardians seemed to be less afraid when they had BA as back-up, and because of that the legal guardians were acting calmer towards their adolescents. Participants also described BA as offering a break in worn out relationships between the adolescents and the rest of the family. They said that BA was reducing stress for all parties as they were able to exhale on their own—and through that participants said that BA offered some level of healing.


*“For him and for his parents and siblings, it meant that the stress subsided. Or how should I explain it? Well, they were extremely worried. And so was he. He felt really bad. But once he made time/for BA/, (…) That’s when/parents and siblings/knew that: okay, now he’s there. Now nothing will happen. Now his thoughts won’t win.”*

*(interview 1)*


Participants described that at times when the adolescents could be afraid to talk about suicide, they were able to do so during BA and through that cool down the entire situation. They said that the supportive meetings were the best part of BA, where the adolescents talked, and the HCPs listened and tried to guide them into managing everyday life and school. Participants also said that the booked meetings could vary in usefulness and content depending on the adolescents’ needs at the moment. Overall, BA was described as reducing complexity and stress in the family system, when offering adolescents the opportunity to collect their thoughts and regain energy.


*“Instead of being so dramatic where they come in with a lot of noise and fuss, there’s an outlet and a space for them to… uh… where they can pull away and land a little. And that they… do it, with staff, that is, healthcare staff who are emotionally, how should I put it, cool.”*

*(interview 12)*


Participants shared experiences of how BA also offered relief to outpatient HCPs with a high workload and emotional engagement in their job with adolescents at risk of suicide. They described being able to suggest BA to adolescents—and how they felt safer when adolescents who had self-harmed during compulsory admissions before, had access to BA. Outpatient HCPs described that BA had made it easier for them to work with and meet the families.

### 3.2. Work Processes Needing Communication and Adaptation

#### 3.2.1. Being Grounded in Leadership and Intense Outpatient Care

Participants shared experiences of how the successful implementation and resilience of BA over the years was a direct result of clear and strong clinical leadership. The leadership standing behind the BA initiative was emphasized as crucial to making sure BA did not gradually erode or turn into emergency care. The consideration of suicide risk in the target group was one aspect where the clinical leadership was brought forward as essential in the implementation of BA.


*“We discussed a lot the wording around suicide risk assessment. (…) In connection with BA, we assessed the risk as lower than on regular admission or in the case of no BA at all./The chief medical officer/was such an important person as a manager who gave us permission to do this. (…) To go on training and provide space for/BA/. And/the chief medical officer/was also involved in these discussions.”*

*(interview 14)*


Another important factor related to leadership, shared by the participants, was the establishment of a BA coordinator position, at the time of the study held by a licensed practical nurse specialized in psychiatric care and experienced in working with adolescents with self-harm at risk of suicide. The same person had held the position since the implementation of BA seven years earlier and was described as a hub and a driving force, keeping the oversight to make sure BA was kept according to the manual.


*“If one aim is for adolescents to be able to strengthen their autonomy and take greater responsibility for their well-being. (…) Then there is a need for someone to maintain the structure during admissions. (…) I think that/the BA coordinator/does that very well.”*

*(interview 2)*


Participants shared experiences of how the coordinator held regular information meetings regarding BA history, status, negative and positive feedback from adolescents. This was said to be important for new colleagues and for continuous repetition, giving examples of how HCPs who had met the adolescents in different situations had been unknowledgeable about BA, such as a psychiatrist having inappropriately suggested BA in the emergency room or substitutes having been unaware of the content of the contract, trying to activate adolescents who had written that they wanted to be left alone to read. Participants said that this pointed towards the need for those being responsible for BA to continuously and relentlessly provide information about BA. They also talked about the importance of the substitute coordinator covering up for the coordinator during holiday seasons.

Although BA was provided in inpatient care, participants emphasized the important role of outpatient care in relation to BA. Ongoing outpatient care was described as essential, where BA combined with, for example, DBT was experienced to provide continuity and a tool facilitating work on agency through skills training. Participants from both inpatient and outpatient care agreed that outpatient HCPs had a key role in supporting the adolescents to, through safety and interplay, find the drive and confidence to use BA. Participants emphasized that outpatient care, being responsible for ongoing treatments, had a say in when the timing for initiating BA could be beneficial or not and stressed the importance of continued outpatient care treatment during BA. The writing of the BA contract taking place in the outpatient care setting was brought forward as an important signal that treatment was grounded in outpatient care.


*“There is a symbolism in it, that this is a tool used in inpatient care. But that it is a complement to regular treatment. (…) It is not, like, instead of treatment in outpatient care.”*

*(interview 2)*


Participants described that an effect of the emphasis on outpatient care in relation to BA was strengthened self-confidence among outpatient HCPs and experiences of increased collaboration between outpatient and inpatient care in relation to adolescents with self-harm at risk for suicide. Again, the BA coordinator role was brought forward as essential, as outpatient care HCPs said that they felt like they could easily reach inpatient care and book meetings via the coordinator.

#### 3.2.2. Working on Clear Communication

Participants shared examples of how they over the years had clarified their way of communicating with adolescents and families around BA. The contract negotiation was described as important, based on the BA manual, mutual agreement and knowledge about BA. Consensus around BA between outpatient and inpatient care was brought forward as essential.

Participants shared examples of how they had improved in informing adolescents and legal guardians about the implications of BA and stressing the adolescents’ ownership in relation to BA. They gave examples of how they described BA to adolescents as an alternative to using the knife when boiling over, not feeling understood or getting stuck. Participants described how they put effort into talking about the individual triggers and at what points or situations it could be beneficial for the adolescent to use BA. Participants said that adolescents usually were attracted by BA due to the possibility to be admitted without legal guardians, under individually adapted circumstances. Booking a meeting to discuss BA was a possibility for signing a contract. Participants described that adolescents were attracted by BA as an individual offer.


*“They’re picking up on it, especially the fact that it’s something for them. That it’s not about the parents. And that they… like they like it to be tailored a little bit to their liking. (…) I think I’m planting a seed somewhere so they can start thinking about it. And then/the BA-coordinator/comes and fills in. And usually, yes, in the vast majority of cases, they buy it.”*

*(interview 12)*


To help adolescents to dare to try BA, participants pointed out that assistance from outpatient HCPs who knew the adolescents well was required, reminding adolescents who were hesitant or frightened to take the initiative to BA. Outpatient HCPs also described how they worked to facilitate and encourage BA usage by calming down legal guardians. Helping legal guardians to let go of their control and motivate them to encourage their adolescents to use BA was described as a crucial legal guardian role in relation to BA.

Participants described how contract negotiations or renewals and HCP meetings over the years had been characterized by increased emphasis on how important it was for HCPs to validate adolescents when they were using BA. Sticking to the biannual contract negotiations was experienced as important because a lot happened in the adolescents’ lives, in terms of, for example, family situation, living conditions and healthcare. Participants said that HCPs working on the BA unit needed to be constantly aware of the fact that adolescents on BA were in the process of dealing with themselves, and that fighting for autonomy would be hard when not feeling well, which could be a hurdle to use BA. Participants warned about the risks of adolescents on BA being viewed as ‘easier patients’ which could affect the approach and attention to them negatively.


*“The approach is crucial and to see them so that they don’t just become wallpaper flowers.”*

*(interview 13)*


Participants talked about the value of complementing adolescents for using BA and acknowledging that HCPs knew that the adolescents were doing their best. They also described experiences of the importance of non-verbal communication with adolescents self-admitting to BA, being perceived as extra sensitive since BA was on the initiative of the adolescents. An example was giving adolescents a high five or a hug when meeting them at the unit. Compared to when admissions are decided by healthcare, participants described that BA could be quite difficult to get started with and would require a lot of encouragement. Autonomy, which was the goal of BA, required well-trained HCPs in inpatient and outpatient care, to ensure that HCPs approached adolescents on BA with respect and consideration, while closely following the set-up. The supportive meetings with a clear purpose were highlighted as key in relation to communication within BA.


*“It is not the purpose that nursing staff (…) should have almost psychological conversations (…) Then you have to refer to the regular/outpatient care/contact. (…) Focus should be on the here and now. Radiate calmness in response to the entire situation.”*

*(interview 2)*


#### 3.2.3. Being Limited by an Inflexible Care Setting

Participants shared reflections on the fact that BA was provided at a unit for children with eating disorders. Some thought that the unit for eating disorder was calmer than the alternatives, while others thought that this arrangement was beneficial for neither the adolescents on BA nor those admitted for eating disorder. Participants said that the arrangement affected the care environment for adolescents on BA in a negative way being characterized by strict meal plans, anxiety in relation to meals and the risk of triggering eating problems in adolescents on BA. Some participants said that they had experienced being unable to meet the social needs of adolescents on BA due to the rules, routines and work related to meals for the others being admitted at the unit.


*“There is almost a ‘by the hour’ anxiety among the eating disorder patients. Many of these girls/on BA/also have an eating problem that triggers them. (…) So many of them… have asked to be in another unit. Have been rejected at the moment. And some have even… not wanted to have BA due to this.”*

*(interview 13)*


Participants said that unlocked doors would be ideal on BA but that this was impossible for a pediatric unit due to the legal aspects and because BA was situated in a unit with other others needing locked doors. However, they also said that locked doors were perceived as beneficial as the adolescents needed to speak to someone before going outside even for a smoke. It provided HCPs with a sense of security as they were always speaking to the adolescents before they were leaving. Participants said that perhaps this reduced risks with adolescents wanting to leave due to acute crisis and suicidality.

Participants asked for increased responsibility within the CAP system to proactively secure access to BA for young adults in need, sharing examples of abrupt changes where BA had been terminated on the day the adolescent turned eighteen. Considering the risk of overthrowing previous results due to care disruption or even risking suicide, participants pointed toward the need to improve continuity between BA for adolescents and adult psychiatry. Participants suggested that the coordinator, being responsible for contract renewal in CAP, could be a resource in this perspective.


*“In general, the bridging between CAP and adult psychiatry is poor. (…) I think it should be possible to guide from BA/within CAP/to BA for adults.”*

*(interview 13)*


#### 3.2.4. Lacking Legal Guardian Participation

Participants stressed the need to expand on the work with BA in relation to the whole family and increase the involvement of legal guardians, so that BA did not become something like ‘taking the car into service’ (Interview 5). When the adolescent called regarding BA and legal guardians were following them to the door, the legal guardians, who otherwise were available 24/7 for their adolescent, tended to be almost pushed away and uninvolved, according to participants, which understandably could make them nervous. Participants said that the lack of involvement of legal guardians was a weak spot of BA and a result of BA being originally developed for adults.


*“Adolescents are free to leave./Legal guardians/don’t have control over them in the same way as if they were on an emergency admission.”*

*(interview 2)*



*“We have inpatient care which states that parents should be present and involved and so on. And then we have/BA/which is largely structured as if it was for adults. (…) How parents should be able to be involved in BA is not in focus.”*

*(interview 6)*


Participants described legal guardians who were living through difficult times, struggling to keep up with their child’s emotional turmoil and dangerous behaviors, and not seldom while also lacking a constructive dialogue with their child. They described how they worked to support legal guardians to manage expectations of BA, trying to make sure that legal guardians understood the BA concept better. Participants said that they had experienced that legal guardians’ stress and nervousness regarding BA disappeared gradually but also said that they felt humble towards feelings of lacking control among legal guardians. Involving legal guardians more in BA was brought forward as crucial because for one thing the feelings of the legal guardians, for example, nervousness, would ultimately affect their child. By trying to hold the legal guardians’ anxiety, the legal guardians could be more able to relax during BA. Participants described how they were following up with legal guardians when their adolescent was on BA to make sure the legal guardians did not feel abandoned, which in turn would make it easier for the adolescents to use BA.


*“To reassure/the legal guardians/about/BA/. That it’s okay that they don’t have control in there. (…) So, it’s probably about holding the parents. (…) And meet them in their anxiety so that the adolescents don’t have to.”*

*(interview 12)*


Participants suggested increased structured involvement of legal guardians in BA, such as a family meeting at discharge, to provide legal guardians with information on how they may support an adolescent in their skills. They said this would be especially valuable in cases where the adolescent may not be on speaking terms with their legal guardians.

## 4. Discussion

### 4.1. Discussion of Results

Results indicated BA being perceived as an intelligent model of care adapted to the needs of adolescents with self-harm at risk of suicide. Several parts of the results are recognized from other studies on BA. Firstly, the perceived value of having access to BA for safety and trusting in its availability has been reported in prior research on BA ([34]). This is also evident in our study, exemplified by HCPs experiencing adolescents who had kept renewing their contract without actually self-admitting to BA. The value of BA being brief, guaranteeing access, promoting autonomy and offering relief to the system has been repeatedly emphasized in prior research ([7]; [11]; [16]; [17]; [34]).

Managing school was brought forward as important for the adolescents and a common goal of BA in the contracts, according to experiences. To the best of our knowledge nothing similar, such as goals of being able to manage work, has been reported in research on BA for adults. However, adolescents’ focus on school can be recognized from interviews with adolescents who have suggested that it would be beneficial to provide more information about BA to schools ([19]). Perhaps an interesting follow-up study could be to review BA contracts within CAP and adult psychiatry to analyze and compare the content of goals and individual needs during BA to gain additional knowledge.

The perceived risks raised by participants in this study of overriding legal guardians in relation to BA has been confirmed by experiences among legal guardians sometimes feeling out of control and scared during BA as adolescents are free to leave the unit unaccompanied and fearing that adolescents are not taken care of during BA ([11]). On the other hand, research has indicated that adolescents with self-harm at risk of suicide highly appreciate being able to self-admit to BA without accompanying legal guardians, enabling them to focus on their own well-being rather than holding back own emotions to protect legal guardians ([19]). Our study indicated that the HCP, being a third party in relation to adolescents and legal guardians, indeed did experience a need—and in some cases an ability—to serve as a mediator helping the legal guardians feel safe in relation to BA, so that the adolescents could use BA according to their needs. There may be a need to take the whole family into more consideration in BA for adolescents with self-harm at risk of suicide, which has also been indicated by previous research based on experiences among family members ([11]). According to a previous study, HCPs describe how they try to involve legal guardians without taking away the adolescent’s voice, finding it important to balance this with a focus on the adolescent’s recovery ([22]). When legal guardians comprehend the essentials of BA, they can probably improve the support of their contracted adolescents. Well-informed legal guardians can strengthen family cohesion and support the adolescents’ psychological development ([28]) and most likely address early signs of crisis reducing the risk of escalation and readmissions. Yet, the adolescent’s autonomy must be obeyed, i.e., legal guardian involvement must be weighed with the growing adolescent’s wishes and capacity ([10]). Some families are suffering from severe dysfunctional relations, i.e., legal guardian involvement must be adjusted individually ([31]) and not assumed to be of support for all adolescents with a BA contract. Our results point toward the value of informing and preparing legal guardians more in relation to BA to make sure they understand the BA concept and their adolescent’s specific needs and development, thereby meeting the needs of the legal guardians and supporting recovery as a relational process ([16]). Additional work to consider the legal guardians could be necessary as part of the continued adoption of BA into CAP, being originally developed for adults, as was raised in prior research on HCPs experiences of BA in CAP ([23]).

Challenges related to BA usage, such as the risk of being triggered when BA is provided together with emergency admissions within the same unit, are recognized from earlier studies ([16]; [18]). The mixing of BA with other admissions may contribute to HCPs viewing adolescents on BA as ‘easy’, as was mentioned in our study, which could make working on autonomy difficult if adolescents are feeling vulnerable. Some may struggle to dare to seek BA or ask for help during BA. HCPs working with BA will need to stay aware of common hurdles for help-seeking among adolescents with self-harm, such as fear of negative reactions and of being perceived as seeking attention ([26]). Situated at a unit with locked doors because BA was mixed with emergency admission provided participants with a sense of safety. Although understandable, this could raise signals of distrust in adolescents’ abilities and go against important self-perceived needs for recovery among adolescents with suicidal behavior, including sensing control, independence, and autonomy ([21]), which is at the core of BA ([12]).

The key role of outpatient care and BA being viewed as a tool for outpatient care rather than a treatment clearly contributed to the experiences of how BA linked outpatient and inpatient CAP. This phenomenon is recognized from HCPs experiences in adult BA ([15]; [17]), fostering mutual respect and visualizing the different roles of HCPs involved in BA work. The motivational work performed in outpatient care highlighted as key for BA to work as intended is also reported on within the adult version of BA, referring to BA as a natural part of a structured crisis plan worked out between outpatient HCPs and individuals with a BA contract ([17]).

The significant role of clinical leadership being actively participating in practical discussions and investing in training and human resources appears crucial given that BA is characterized by withdrawal of physician control and with licensed practical nurses as the primary contacts for BA ([9]). The successful appointment of a strong BA coordinator, in this case described by all participants as being the right person in the right place could be a potential vulnerability should the person leave the position, although a back-up coordinator being in place reduces the risk.

Bridging from CAP to adult psychiatry is a general issue raised within psychiatry ([14]), also raised in our study in relation to BA. Contrasting transfer of care, indicating interruption, with transition of care, the latter implying care continuity which is parallel with planning meetings and information transfer ([24]) may be beneficial to consider in relation to our results. While some participants in our study brought forward critiques of the move from CAP to adult psychiatry, including experiences of an abrupt end to BA, a member of the research group performing this study is aware of hospitals in the region where CAP is invited to attend the first BA contract negotiation in adult psychiatry when an adolescent with continued need of BA has turned eighteen. Sharing procedures between hospitals in the region and having informational meetings on BA with representatives from both CAP and adult psychiatry could contribute to increased harmonization through the sharing of good examples.

### 4.2. Trustworthiness

Several measures were taken to achieve trustworthiness according to the chosen methodology ([4]). Activities to achieve credibility included putting effort into deciding on how to reach participants with different experiences, as we wished to include HCPs from both inpatient and outpatient care and without limitations in professional background or years of experience in psychiatry. All employees in CAP care in the region were contacted to allow HCPs themselves to decide if they had experience of BA. This resulted in a variation in age, gender, professional background, and role in relation to BA for adolescents, which shed light on BA from various aspects. BA being a complex intervention ([29]) points towards the importance of different stakeholder perspectives in evaluation, where our study adds to the previous research where adolescents and legal guardians have been interviewed ([11]; [19]) Data analysis was based on collaboration and dialogue between two authors (RL and KL), who discussed subjective interpretations, coding and themes for confirmability. Results were discussed between all authors, and representative quotes from the interviews were used to demonstrate sub-themes and themes. In relation to dependability data was collected over a brief period supporting consistency. Digital interviews were somewhat shorter than interviews performed face to face, which may have reduced the amount of data. We aspired to provide a transparent description of the setting, the study process and findings for transferability to the reader. Finally, positioning us as researchers, we acknowledge that our backgrounds have shaped study design and our understanding of the results. All authors had extensive experience of working with BA and/or doing research on BA, and different backgrounds (educated in public health, nursing, and psychiatry) providing dynamic discussions in the process of this research.

## 5. Conclusions

In this study the most pivotal aspects to make the concept of BA work for adolescents have been identified by HCPs working with BA. The results support previous research: BA seems to suit adolescents with self-harm at risk of suicide very well. Dedicated and supportive clinical leadership allowing for continuous training in BA seems to be a vital prerequisite for long-term continuation. Being grounded in outpatient care, BA provides an extension of their toolbox. A reliable and committed BA coordinator seems to be essential, and this position needs a back-up to ensure that the intervention can be up and running regardless of changes in staffing. By 2024, BA for adolescents was implemented in only 3 of the 21 Swedish regions ([30]). Our results can support future implementations and might add guidance around pitfalls to avoid.

Areas for future improvement highlighted in this study are increased involvement of legal guardians in an amount suitable for adolescents, striving for a care environment appropriate for BA, preferably by allowing for a unit exclusively for BA, and finally to smoothen the transition from adolescent to adult psychiatry for those with BA.

## Figures and Tables

**Table 1 behavsci-15-01210-t001:** Healthcare professionals’ (HCPs) roles and responsibilities in the delivery of Brief Admission by Self-referral (BA) for adolescents.

BA Coordinator
General communication regarding BA Hold recurrent meetings to update inpatient HCPs about BA, report and discuss feedback from adolescents. Encourage inpatient HCPs to present a professional welcome to adolescents on BA and make sure that basic elements of BA are kept. Inform outpatient HCPs about adjustments, e.g., changes in HCPs approach against adolescents, treatment plans and routines. Educated new HCPs into the concept. Keep track on adolescents who might be relevant for a BA contract. Check the inclusion criteria and discuss them with the designated senior psychiatrist before proceeding with the process with BA candidates. Keep daily (Monday to Friday) track on notifications regarding BA candidates initiated by outpatient HCPs. Keep daily track on admitted potential BA candidates during the morning rounds at the emergency unit (Monday to Friday) and on adolescents in psychiatric emergency admission. Contract negotiations Approach adolescents with presence and seriousness but without being too solemn, encourage a laugh and set a tone for positive expectations. Lead the negotiations. Discuss and formulate individual aspects and wishes during a BA admission into the contract template, e.g., triggers, treatment, and desired pronoun when being addressed. Mention that a contract evaluation with potential contract renewal will take place every six months, including feedback, repetition, and alliance building. Explain that BA due to overcrowding temporarily may be inaccessible, and that inpatient care HCPs on these occasions are available for support on the phone and that it is possible for the adolescent to try again the next day. Explain that BA does not affect access to the usual emergency care which is open 24/7 when this is needed. Sign the contract together with the adolescent, at least one legal guardian, and outpatient HCPs. Register and scan the contract into the adolescent’s medical records. During BA admission Now and then suggest a walk in the park with the adolescents and propose a joint meal in the unit as the adolescents are often admitted without legal guardians. Perform daily checks on the structured care plans based on participation, and act if, for example, time for discharge is missing. Talk to the adolescent and contact a legal guardian if there are any indications of mistakes or discontent. Ensure that there are always available rooms for BA and assist to transfer children on emergency admission from a BA room if they have been temporarily admitted there overnight. Termination of BA contract Terminating the BA contract, for example, when the adolescent wishes so or when they turn eighteen and make a notification in the adolescent’s medical records.
Inpatient Care HCPs at the Unit Offering BA (Registered Nurses and Licensed Practical Nurses)
Answer the phone, welcome adolescents and set a minor agenda regarding time for arrival. Give a heads-up to other HCPs at the unit about the adolescent and write a note in medical records. When the adolescent arrives, together with the adolescent read through the BA contract and write a treatment plan with goals for the current admission together with the adolescent and a legal guardian, either present on-site or via phone. Review unit routines and show the adolescent to their room. Book supportive meetings concerning the here and now situation, up to two per day, 15–20 min each, with the adolescent on BA. Deliver prescribed medications to adolescents on BA (registered nurses only). Introduce and supervise new HCPs into the BA concept. Discharge the adolescent in collaboration with a legal guardian according to the treatment plan.
Outpatient Care HCPs
Contact the BA coordinator regarding BA candidates. After coordination with the BA coordinator inform adolescents and legal guardians about the BA concept. Support the adolescent and legal guardian during contract negotiations. Ensure that adolescents maintain and, in some cases, improve their anxiety management skills. Together with the BA coordinator be responsible for contacting adolescent and legal guardians when it is time for a contract evaluation and potential contract renewal.
Designated Senior Psychiatrist in Inpatient Care
Overall medically responsible for the adolescent during BA. Together with the BA coordinator discuss presumable BA candidates. Together with the BA coordinator discuss problems as they arise. Write a standardized discharge summary in the adolescent’s medical records after discharge.
Other HCPs not Directly Involved in BA
Doctor on call—prescribe medicines in the medical records.

**Table 2 behavsci-15-01210-t002:** Participant characteristics (n = 14).

Gender	5 males/9 females
Age, range (median)	35–67 (60)
Experience of working in Child and Adolescent Psychiatry	
Participants with experience from outpatient care	9
Participants with experiences from inpatient care	6
Years working in CAP, range (median)	3–40 (20)
Profession	
Licensed practical nurse	5
Psychologist	3
Registered nurse	2
Social worker	2
Behavioral scientist	1
Psychiatrist	1

**Table 3 behavsci-15-01210-t003:** Themes and subthemes.

Theme	Caretaking Without Taking Over	Work Processes Needing Communication and Adaptation
Subtheme	Promoting mental growth and agency	Being grounded in leadership and intense outpatient care
	Being brief	Working on clear communication
	Granting access	Being limited by an inflexible care setting
	Offering relief to the system	Lacking legal guardian participation

## Data Availability

Interviews supporting the findings of this study are not publicly available due to privacy.

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
