# Peer review of "Healthcare Professionals’ Experiences of Brief Admission by Self-Referral for Adolescents with Self-Harm at Risk of Suicide—A Qualitative Interview Study"

_behavsci, 2025, doi:10.3390/bs15091210_

Round 1
Reviewer 1 Report
Comments and Suggestions for Authors
I read your work with great interest. You explore a crucial and timely issue in Child and Adolescent Psychiatry: ensuring continuity of care for young people and preventing disruptive behaviors, which relies heavily on strong collaboration within care networks. I find that the chosen approach is effective, and the inclusion of caregivers’ perspectives adds meaningful depth to the study. Your research is thoughtfully designed, clearly presented, and makes a valuable contribution to the field. You also provide a balanced discussion by acknowledging both the strengths and limitations of their work. Thank you !
Author Response
Author's Reply to the Review Report (Reviewer 1)
Comment 1: I read your work with great interest. You explore a crucial and timely issue in Child and Adolescent Psychiatry: ensuring continuity of care for young people and preventing disruptive behaviors, which relies heavily on strong collaboration within care networks. I find that the chosen approach is effective, and the inclusion of caregivers’ perspectives adds meaningful depth to the study. Your research is thoughtfully designed, clearly presented, and makes a valuable contribution to the field. You also provide a balanced discussion by acknowledging both the strengths and limitations of their work. Thank you !
Response 1: We very much appreciate your review of our manuscript, and the feedback provided. Thank you!
Reviewer 2 Report
Comments and Suggestions for Authors
Thank you for the opportunity to review the manuscript entitled ‘Healthcare Professionals’ Experiences of Brief Admission by Self-referral for Adolescents with Self-harm at Risk for Suicide – a Qualitative Interview Study. ID: behavsci-3746060
This is a well-written paper, and the authors highlight aspects of BA in relation to children and adolescents at risk of self-harm or suicide. This is a timely research area, as BA is mainly offered to adults and not to the same extent to children and adolescents. The study contributes valuable knowledge about the implementation of BA for a younger group and the factors that are important for making BA as accessible and effective as possible.
I have only a few minor points that I noticed and that could be clarified in the manuscript.
Keywords: The keywords would be clearer if arranged in alphabetical order. The reference “Moberg” on line 590 is missing a publication year — please check this reference in the reference list.
Background: Well written; it provides a solid presentation of the target group (children and adolescents) in relation to the issue. The manuscript offers a detailed and thorough description of BA and the responsibilities of the various professional groups involved. Relevant articles in the field are appropriately cited.
Method: Well written, with a clear description of the data collection process, and the context is presented effectively.
Results: The results are well written and highlight both the positive experiences and the challenges of BA. The significant role of clinical leadership is an important finding and should not be underestimated. The study conveys that BA requires organisation, committed leaders, and dedicated staff in order to work in practice. The inclusion of quotations that strengthen the results is a strong aspect.
Analysis: The analysis is somewhat superficial and simplified. A meaning unit can be understood as words, sentences, or paragraphs that are connected to each other by their content and context. The analysis could be developed further by describing how the meaning units were identified and what the authors did in the subsequent analysis steps. If the analysis procedure was a dynamic process, this should be made explicit. The authors analyse the data at both a manifest and a latent level — it would be helpful to provide a bit more information about these approaches.
Otherwise, I find that the conclusions drawn from the study are appropriate and relevant. The limitations of the work are acknowledged. The text accurately and explicitly reports the findings as evidenced by the data.
Author Response
Author's Reply to the Review Report (Reviewer 2)
Comment 1: Thank you for the opportunity to review the manuscript entitled ‘Healthcare Professionals’ Experiences of Brief Admission by Self-referral for Adolescents with Self-harm at Risk for Suicide – a Qualitative Interview Study. ID: behavsci-3746060. This is a well-written paper, and the authors highlight aspects of BA in relation to children and adolescents at risk of self-harm or suicide. This is a timely research area, as BA is mainly offered to adults and not to the same extent to children and adolescents. The study contributes valuable knowledge about the implementation of BA for a younger group and the factors that are important for making BA as accessible and effective as possible. I have only a few minor points that I noticed and that could be clarified in the manuscript.
Response 1: Thank you! We very much appreciate your positive response and constructive feedback.
Comment 2: Keywords: The keywords would be clearer if arranged in alphabetical order.
Response 2: We agree and have arranged the keywords accordingly, see tracked changes on page 1, rows 31-34.
Comment 3: The reference “Moberg” on line 590 is missing a publication year — please check this reference in the reference list.
Response 3: Thank you for bringing this to our attention. We have checked and corrected the reference by adding the publication year in text (page 15, row 600) and corrected the reference in the reference list (reference number 21 on page 19, row 784).
Comment 4: Background: Well written; it provides a solid presentation of the target group (children and adolescents) in relation to the issue. The manuscript offers a detailed and thorough description of BA and the responsibilities of the various professional groups involved. Relevant articles in the field are appropriately cited. Method: Well written, with a clear description of the data collection process, and the context is presented effectively. Results: The results are well written and highlight both the positive experiences and the challenges of BA. The significant role of clinical leadership is an important finding and should not be underestimated. The study conveys that BA requires organisation, committed leaders, and dedicated staff in order to work in practice. The inclusion of quotations that strengthen the results is a strong aspect.
Response 4: Thank you, we appreciate your feedback!
Comment 5: Analysis: The analysis is somewhat superficial and simplified. A meaning unit can be understood as words, sentences, or paragraphs that are connected to each other by their content and context. The analysis could be developed further by describing how the meaning units were identified and what the authors did in the subsequent analysis steps. If the analysis procedure was a dynamic process, this should be made explicit. The authors analyse the data at both a manifest and a latent level — it would be helpful to provide a bit more information about these approaches.
Response 5: We value your feedback, and we agree that these aspects could be further clarified. We have therefore expanded on the following points in the section on Analysis (see tracked changes on page 6, rows 183-202):
- provided more details regarding how meaning units were identified
- clarified who did what in the analytical process
- stated that the analysis was a dynamic process
- added more detail regarding manifest and latent level
We think that the added text has improved the Analysis section so that it provides increased transparency to the reader regarding how the analysis was performed.
Comment 6: Otherwise, I find that the conclusions drawn from the study are appropriate and relevant. The limitations of the work are acknowledged. The text accurately and explicitly reports the findings as evidenced by the data.
Response 6: Your feedback has contributed to improving our manuscript further and we very much appreciate your valuable and thorough review. Thank you!